# Role of sphingosine kinase and sphingosine-1-phosphate receptor in the liver pathology of mice infected with *Plasmodium berghei* ANKA

**Tachpon Techarang[1], Pitchanee Jariyapong[1], Chuchard Punsawad** [1,2]*

**1** Department of Medical Science, School of Medicine, Walailak University, Nakhon Si Thammarat, Thailand,
**2** Research Center in Tropical Pathobiology, Walailak University, Nakhon Si Thammarat, Thailand

* chuchard.pu@wu.ac.th

**Data Availability Statement:** All relevant data are within the paper.

**Funding:** This study was supported by Walailak University, Thailand. The funders had no role in

## Abstract

Decreased serum sphingosine 1-phosphate (S1P) has been reported in severe malaria patients, but the expression of receptors and enzymes associated with S1P has not been investigated in the liver of malaria patients. Therefore, this study aimed to investigate the expression of sphingosine kinase (SphK) and S1P receptors (S1PRs) in the liver of malaria-infected mice. C57BL/6 male mice were divided into a control group (n = 10) and a *Plasmodium berghei* (PbA)-infected group (n = 10). Mice in the malaria group were intraperitoneally injected with 1×10$^6$ *P. berghei* ANKA-infected red blood cells, whereas control mice were intraperitoneally injected with normal saline. Liver tissues were collected on Day 13 of the experiment to evaluate histopathological changes by hematoxylin and eosin staining and to investigate SphK and S1PR expression by immunohistochemistry and real-time PCR. Histological examination of liver tissues from the PbA-infected group revealed sinusoidal dilatation, hemozoin deposition, portal tract inflammation and apoptotic hepatocytes, which were absent in the control group. Immunohistochemical staining showed significant increases in the expression of SphK1 and SphK2 and significant decreases in the expression of S1PR1, S1PR2, and S1PR3 in the endothelium, hepatocytes, and Kupffer cells in liver tissue from the PbA-infected group compared with the control group. Real-time PCR analysis showed the upregulation of SphK1 and the downregulation of S1PR1, S1PR2, and S1PR3 in the liver in the PbA-infected group compared with the control group. In conclusion, this study demonstrates for the first time that SphK1 mRNA expression is upregulated and that S1PR1, S1PR2, and S1PR3 expression is decreased in the liver tissue of PbA-infected mice. Our findings suggest that the decreased levels of S1PR1, S1PR2, and S1PR3 might play an important role in liver injury during malaria infection.

## Introduction

Malaria is a life-threatening infectious disease caused by *Plasmodium* parasites that remains a cause of morbidity and mortality in humans; there were 229 million clinical cases in 2019 and 409,000 deaths worldwide, of which 67% were children under 5 years of age [1]. *Plasmodium*

study design, data collection and analysis, decision to publish, or preparation of the manuscript.

**Competing interests:** The authors have declared that no competing interests exist.

parasites are typically transmitted to humans by the bite of an infected female *Anopheles* mosquito. *Plasmodium* parasites travel to the liver, where they mature and reproduce [2]. There are several severe complications of malaria, such as cerebral malaria, acute lung injury/acute respiratory distress syndrome, acute kidney injury, hypoglycemia, lactic acidosis, severe anemia and acute liver failure [3]. The liver is the first important organ affected during the hepatic stage of infection and is the site in which malaria sporozoites develop into merozoites. Protozoan parasites damage hepatocytes in all stages of the malaria parasite life cycle, especially in the hepatic stage [4]. In addition, liver damage may be caused by the binding of infected red blood cells to endothelial cells, leading to the obstruction of hepatic sinusoids and intrahepatic blood flow. It has been reported that histopathological changes in liver tissue during malaria infection are characterized by Kupffer cell hyperplasia, abundant hemozoin pigment in hyperplastic Kupffer cells and hepatocytes, inflammatory cell infiltration in the portal tract, (especially neutrophils and monocytes), fatty liver changes, liver fibrosis, cholestasis, bile stasis, granulomatous lesions and malarial nodules [4–6].

Sphingosine 1-phosphate (S1P) is a lysophospholipid mediator that can stimulate a wide variety of intracellular and extracellular responses, such as cell proliferation, differentiation, migration, contraction, and survival, as well as the immune response [7–10]. S1P is released after the phosphorylation of sphingosine by sphingosine kinase (SphK) [11]. Indeed, previous studies revealed that S1P signaling is mediated by a family of five high-affinity G protein-coupled receptors, including S1PR1, S1PR2, S1PR3, S1PR4, and S1PR5 [11–13]. However, only three receptors for S1P are expressed in the liver: S1PR1, S1PR2, and S1PR3. SphKs and S1PRs have been shown to have novel therapeutic potential in the treatment of cancer, gastrointestinal disease, obesity, and diabetes mellitus and its associated complications [9, 10, 14, 15]. In addition, the overexpression of SphK1 in macrophages conferred resistance to mycobacterial infection [16]. In the context of malaria, a recent investigation showed reduced serum levels of S1P in patients with severe malaria [17, 18]. Therefore, we hypothesized that SphK/S1PR plays a role in the severity of malaria and liver damage. However, there have been no reports on the expression of receptors and enzymes associated with S1P in the liver in the context of malaria.

Therefore, this study aimed to investigate histopathological changes and the expression of SphKs (SphK1 and SphK2) and S1P receptors (S1PR1, S1PR2, and S1PR3) in the liver tissue of malaria-infected mice compared with control mice. Understanding the pathophysiological role of the receptors and enzymes associated with S1P may provide new insights into malaria pathogenesis and new therapeutic options for the treatment of malaria infection.

## Methods

### Ethics statement

This study protocol was reviewed and approved by the Animal Ethics Committee of Walailak University (Clearance no. 007/2018). All protocols in this study were conducted in accordance with the relevant guidelines and regulations for using animals in compliance with Animal Research: Reporting of In Vivo Experiments (ARRIVE) guidelines. Animals were anesthetized with pentobarbital to minimize pain and suffering.

### Animal preparation and experimental protocols

Male C57BL/6 mice (6–10 weeks old) were purchased from Nomura Siam International Co., Ltd. (Bangkok, Thailand). The mice were housed at 22–24˚C in standard polycarbonate cages (5 animals per cage) on wood-shaving bedding with free access to food and fresh water and a standard 12 h light/dark cycle. *Plasmodium berghei* ANKA was obtained from BEI Resources (NIAID, NIH: *P. berghei*, Strain ANKA, MRA-311) and was contributed by Thomas F.

McCutchan. Donor mice were inoculated with *P. berghei* ANKA-infected red blood cells, and blood was collected via cardiac puncture into a heparinized vacutainer tube for the induction of malaria in experimental mice [19]. For the malaria-infected group, the mice were intraperitoneally infected with $1\times10^6$ *Plasmodium berghei* ANKA (PbA)-infected red blood cells as previously described [20, 21]. Control mice were intraperitoneally administered 200 μl of saline. After infection, parasitemia levels were monitored daily using Giemsa-stained peripheral blood smears. On the 13[th] day after infection (DAI), the mice were sacrificed under anesthesia with an intraperitoneal injection of pentobarbital (60 mg/kg body weight) to minimize suffering. After opening the abdominal cavity, liver tissues were collected immediately for histopathological, immunohistochemical and real-time PCR analyses. Finally, the mice were intraperitoneally injected with an overdose of pentobarbital (150 mg/kg) for euthanasia.

## Histopathological examination

Liver tissues were fixed in 10% formalin for 24–48 h, embedded in paraffin, sectioned (5 μm thickness) and stained with hematoxylin and eosin as previously described [22]. For histological assessments, alterations in four histological parameters (sinusoidal dilatation, hemozoin deposition, portal tract inflammation and apoptotic hepatocytes) were assessed under a light microscope at high magnification (400×) by two expert pathologists. Based on a previous study with modifications [4], the severity of each parameter was graded using a semiquantitative scoring system, as shown in Table 1. The total histological score was calculated as the sum of the scores for each of the four histological parameters and ranged from 0 to 12 points.

## Immunohistochemistry

Formalin-fixed, paraffin-embedded (FFPE) sections of liver tissue were deparaffinized in xylene and rehydrated in a series of ethanol solutions. FFPE tissue sections were incubated in 3% hydrogen peroxide to quench endogenous peroxidase activity. To improve antigen retrieval, the sections were incubated in Vector Antigen Unmasking Solution (Vector Laboratories Inc., USA) according to the manufacturer's instructions. After being washed in Tris-buffered saline (TBS), the sections were blocked for 30 min in normal goat serum in TBS. FFPE tissue sections were further incubated for 1 h at room temperature with rabbit polyclonal antibodies against SphK1 or SphK2 (1:200; Ab71700 and Ab264042, Abcam, UK) or S1PR1, S1PR2 or S1PR3 (1:200; sc-25489, Santa Cruz Biotechnology, Inc., USA), followed by a 30 min incubation with diluted biotinylated secondary antibodies and a 30 min incubation with VEC-TASTAIN ABC Reagent (Vector Laboratories Inc., USA). Then, all FFPE tissue sections were incubated in DAB reagent (brown chromogen staining; Vector Laboratories Inc., USA), counterstained with hematoxylin, rehydrated, and mounted for light microscopy analysis. Negative controls were prepared in the same manner but without primary antibodies. To evaluate the expression of SphKs (SphK1 and SphK2) and S1P receptors (S1PR1, S1PR2 and S1PR3), the number of positive cells and the intensity of staining in 10 randomly selected microscopic

**Table 1. Grading system for four main histological features to assess the severity of liver injury.**

| Parameters | Grading score | | | |
|---|---|---|---|---|
| | 0 | 1 | 2 | 3 |
| Sinusoidal dilatation | No dilatation | Mild dilatation | Moderate dilatation | Severe dilatation |
| Hemozoin deposition | No deposition | Mild deposition | Moderate deposition | Severe deposition |
| Portal tract inflammation | < 5% of portal tract area | 5–15% of portal tract area | 16–30% of portal tract area | > 30% of portal tract area |
| Hepatocyte apoptosis | No apoptosis | Mild apoptosis | Moderate apoptosis | Severe apoptosis |

fields of each immunostained section were determined at high magnification (400×). The expression of each protein was separately examined in endothelial cells, hepatocytes, and Kupffer cells. The percentage of positively stained cells for each protein marker was calculated by dividing the number of positive cells by the total cell count and multiplying this number by 100. The staining intensity was subjectively scored as follows: 0 = no staining, 1 = weakly positive, 2 = moderately positive, and 3 = strongly positive. Finally, the total score (TS) was calculated by multiplying the percentage of positive cells (%) and staining intensity (I), according to a previous study [23].

### Real-time polymerase chain reaction (real-time PCR)

SphK and S1PR mRNA levels in liver tissue in the control and PbA-infected groups were measured by quantitative real-time PCR. RNA was extracted using TriPure isolation reagent (Roche, Mannheim, Germany) according to the manufacturer's instructions. First strand cDNA was generated using an iScript™ cDNA Synthesis Kit (Bio–Rad, Philadelphia, PA). The reaction was incubated at 25˚C for 5 min, 42˚C for 30 min, and 85˚C for 5 min. cDNA (50 ng) from each sample was subsequently amplified using gene-specific primer sets (**Table 2**). The thermal conditions consisted of initial denaturation at 94˚C for 10 s, followed by 40 cycles of denaturation at 94˚C for 30 s, annealing at 55˚C for 30 s, and elongation at 72˚C for 30 s, and a final elongation step at 72˚C for 5 min. The relative expression of target genes was determined using the $2^{-\Delta\Delta Ct}$ method and beta-actin as a comparator.

### Statistical analysis

Statistical analysis was performed using IBM SPSS version 23.0 (SPSS, IL, USA). The normality of distribution was tested with the Kolmogorov–Smirnov test. Differences between groups (malaria-infected and control groups) were analyzed by a nonparametric Mann–Whitney U test. A p value $< 0.05$ was considered to indicate statistical significance.

## Results

### Histopathological examination

Histopathological analysis of liver tissues from the PbA-infected group (Fig 1A and 1C) showed sinusoidal dilatation, hemozoin deposition, portal tract inflammation and hepatocyte apoptosis compared to those in the control group (Fig 1B and 1C). Semiquantitative analysis

**Table 2. Primer used in this study.**

| Name | Sequence: 5'-3' | Reference |
|---|---|---|
| SphK1-F | GGAGGAGGCAGAGATAACCTT | [24] |
| SphK1-R | GACCCAACTCCTCTGCACACA | [24] |
| SphK2-F | GCCCGAGATGGTCTAGTCT | [24] |
| SphK2-R | GTGGGTAGGTGTAGATGCAGA | [24] |
| S1P1-F | ACTTTGCGAGTGAGCTG | [25] |
| S1P1-R | AGTGAGCCTTCAGTTACAGC | [25] |
| S1P2-F | TTCTGGAGGGTAACACAGTGGT | [25] |
| S1P2-R | ACACCCTTTGTATCAAGTGGCA | [25] |
| S1P3-F | TGGTGTGCGGCTGTCTAGTCAA | [25] |
| S1P3-R | CCCCGTTCTGAAACGACCTG | This study |
| β-actin-F | GTGACGTTGACATCCGTAAA | This study |
| β-actin-R | CTCAGGAGGAGCAATGATCT | This study |

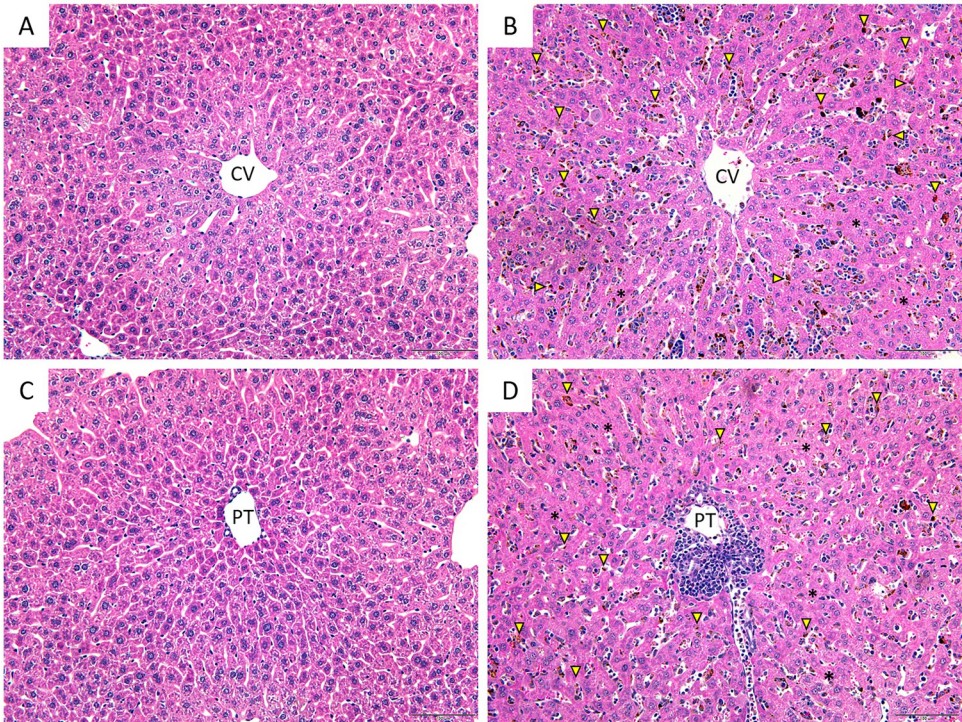

**Fig 1. Histopathological changes in liver tissues from control (A) and PbA-infected mice (B).** The histomorphological changes in PbA-infected mice include hemozoin deposition (indicated by yellow arrowheads) and apoptotic hepatocytes (indicated by black asterisks) around the central vein. Central vein (CV); Portal tract (PT). All images are shown at 200× magnification. Scale bar = 100 μm.

demonstrated that the mean scores of all histological parameters were significantly higher in the PbA-infected group than in the control group (all $p < 0.05$) (Table 3). The histopathological change with the highest mean was hemozoin deposition (2.64 ± 0.13), followed by portal tract inflammation (1.40 ± 0.09).

## SphK and S1PR expression as indicated by immunohistochemical staining

Immunohistochemical analysis of SphK and S1PR expression in liver tissues from control and PbA-infected mice is shown in Fig 2. Positive staining is indicated by brown. The number of positively stained cells and the staining intensity for each protein were independently examined in each individual cell type, including endothelial, epithelial, and Kupffer cells. The mean

**Table 3. Mean scores of four main histological features in tissues from the control (n = 10) and PbA-infected groups (n = 10).**

| Parameters | Score (Mean ± SD) | |
|---|---|---|
| | Control group | PbA-infected group |
| Sinusoidal dilatation | 0.12 ± 0.14 | 1.56 ± 0.13* |
| Hemozoin deposition | 0.00 ± 0.00 | 2.64 ± 0.13* |
| Portal tract inflammation | 0.10 ± 0.10 | 2.20 ± 0.09* |
| Hepatocyte apoptosis | 0.22 ± 0.11 | 1.40 ± 0.09* |
| Total score | 0.11 ± 0.09 | 1.95 ± 0.57* |

*$p < 0.05$ compared with the control group.

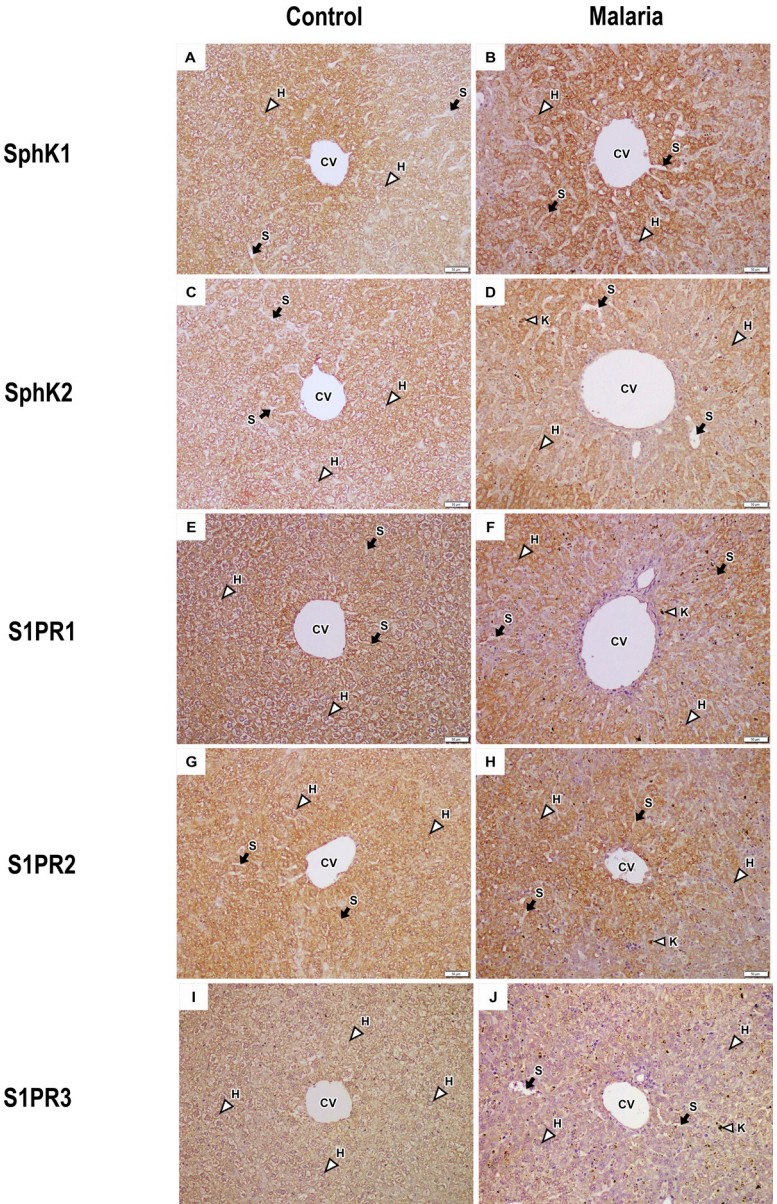

**Fig 2. Immunoperoxidase staining for SphK and S1PR in liver tissues from control (A, C, E, G, I) and PbA-infected mice (B, D, F, H, J).** Hepatocytes (H), Kupffer cells (K), sinusoids (S), and central veins (CVs). All images are shown at 200× magnification. Scale bar = 50 μm.

TS for SphK1 and SphK2 expression was significantly increased in endothelial cells, hepatocytes, and Kupffer cells in liver tissue from PbA-infected mice compared with control mice (p < 0.05). However, the mean TS for S1PR1, S1PR2, and S1PR3 was significantly decreased in endothelial cells, hepatocytes, and Kupffer cells in liver tissue from PbA-infected mice compared with control mice (p < 0.05) (Table 4).

## SphK and S1PR mRNA levels as determined by real-time PCR

The expression of genes encoding enzymes and receptors associated with S31P in liver tissues from control and PbA-infected mice was investigated by real-time PCR (Fig 3). The expression

**Table 4. Mean TS for S1PR and SphK expression in liver tissues from the control (n = 10) and PbA-infected (n = 10) groups.**

| Marker | Group | Endothelium | Hepatocyte | Kupffer cell |
|---|---|---|---|---|
| SphK1 | Control | 48.80 ± 5.79 | 72.70 ± 6.71 | 24.70 ± 2.29 |
| | PbA-infected | 90.90 ± 9.14* | 141.00 ± 10.62* | 58.80 ± 7.16* |
| SphK2 | Control | 35.20 ± 3.97 | 62.10 ± 6.52 | 32.30 ± 3.80 |
| | PbA-infected | 52.30 ± 6.56* | 117.80 ± 10.04* | 60.50 ± 3.04* |
| S1PR1 | Control | 123.80 ± 9.54 | 196.20 ± 12.78 | 27.30 ± 1.94 |
| | PbA-infected | 47.30 ± 3.71* | 95.80 ± 11.00* | 68.40 ± 8.03* |
| S1PR2 | Control | 158.90 ± 14.13 | 221.90 ± 15.08 | 25.40 ± 2.41 |
| | PbA-infected | 50.70 ± 5.57* | 115.40 ± 11.58* | 70.60 ± 7.12* |
| S1PR3 | Control | 91.40 ± 11.11 | 142.10 ± 12.82 | 27.80 ± 2.84 |
| | PbA-infected | 44.40 ± 3.60* | 72.90 ± 8.81* | 56.30 ± 7.05* |

*$p < 0.05$ compared with the control group.

of enzymes responsible for regulating the concentration of S1P, including SphK1 and SphK2, was measured in liver tissue from PbA-infected mice. Our results showed that SphK1 mRNA levels were upregulated approximately 0.5-fold in liver tissue from PbA-infected mice, whereas SphK2 mRNA levels did not change (Fig 3A). In addition, S1PR1 and S1PR2 mRNA levels were decreased in liver tissue from PbA-infected mice compared to control mice, whereas S1PR3 mRNA levels were significantly decreased compared to those in the control group (Fig 3B).

## Discussion

The liver is one of the organs in which infected red blood cells preferentially accumulate to avoid immune system surveillance inside the spleen, and infected red blood cell sequestration within the liver endothelium is now emerging as another factor that promotes liver damage [26–28]. In the present study, histopathological examination demonstrated that liver tissues from PbA-infected mice exhibited sinusoidal dilatation, hemozoin deposition, portal tract inflammation and hepatocyte apoptosis. These findings are consistent with previous studies showing evidence of portal tract inflammation and hemozoin pigment deposition in liver tissue from patients with severe *P. falciparum* malaria and hyperbilirubinemia [4] and from C57BL/6 mice infected with PbA [29], indicating that this murine model could be a potential resource for investigating the pathological mechanisms of liver damage caused by malaria infection.

S1P is a bioactive sphingolipid mediator that functions intracellularly as a second messenger and extracellularly as a ligand for specific G protein-coupled receptors [11–13], affecting processes such as cell proliferation, differentiation, migration, contraction, and survival, as well as inflammation [7–10]. In the context of malaria, protozoan parasite infection can induce the release of proinflammatory cytokines and chemokines, resulting in parasite destruction and an inflammatory response [30, 31]. Our previous study reported that serum S1P concentrations were decreased in malaria patients infected with *P. vivax* and *P. falciparum* and that low levels of S1P were associated with the severity of malaria [18]. However, there have been no reports on the expression of SphK or S1PR in the liver during malaria infection. Herein, we investigated the expression of SphKs (SphK1 and 2) and S1PRs (S1PR1, 2, and 3) in liver tissue from mice infected with PbA. The present study demonstrated SphK and S1PR expression based on positive immunohistochemical staining in three liver cell types, including endothelial cells, hepatocytes, and Kupffer cells. Compared with control mice, PbA-infected mice showed

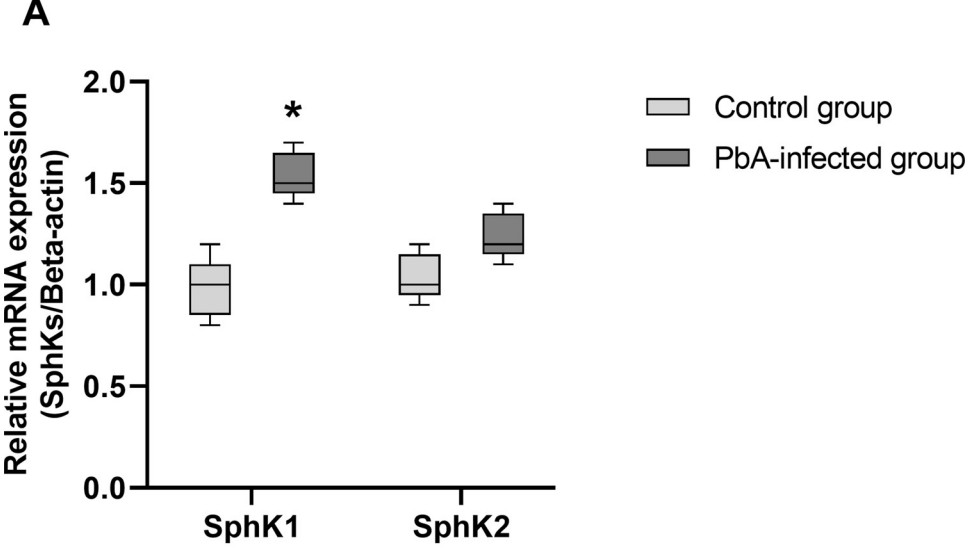

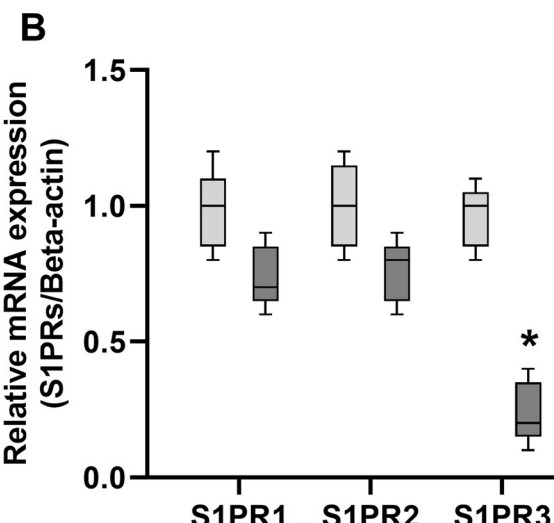

**Fig 3. Relative SphK and S1PR mRNA expression as measured by real-time PCR.** Data are presented as the mean ± SD of triplicate experiments. $^*$p < 0.05 compared with the control group.

a significant increase in the number of cells that were positively stained for both SphK1 and SphK2 and a significant decrease in the expression of all S1PRs in all cell types in liver tissue.

Normally, S1P is located in both intra- and extracellular regions [32]. This protein is exported from cells via transporters such as spinster homolog 2 (Spns2) on the endothelial surface to maintain circulating S1P concentrations [32–34]. S1P is released into the extracellular space after the phosphorylation of sphingosine by SphKs. Therefore, the mRNA expression of receptors and enzymes responsible for S1P signaling was also investigated. The present study revealed that the mRNA expression of S1PRs, especially S1PR3, was downregulated, whereas only SphK1 mRNA was significantly upregulated in PbA-infected mice compared with control

mice. These findings suggest the following. 1) When infected red blood cells rupture during parasite growth and differentiation [2], an important source of circulating S1P is needed due to a lack of cellular organelles [32, 35, 36], and S1P is released into the circulatory system. This protein can bind to plasma components such as high-density lipoprotein (HDL), low-density lipoprotein (LDL), and albumin [37] and thereby decrease S1P activity to stimulate S1PRs on the cell surface, leading to decreases in S1PR activation and expression. 2) Circulating S1P can induce apoptosis through G protein-coupled receptors on the cell surface via caspase-3 [38]. This signaling stimulates the production of inflammatory mediators, the expression of ICAM-1 and VCAM-1 on the cell surface, and the release of Weibel–Palade bodies, which are organelles in endothelial cells that store various proteins involved in inflammation, resulting in an inflammatory response and cell apoptosis [38–40]. 3) The inflammatory response to protozoan parasite infection and hemozoin, a waste product of hemoglobin digestion by the *Plasmodium* parasite, induces leucocyte accumulation. This accumulation leads to the secretion of proinflammatory cytokines, such as tumor necrosis factor alpha (TNF-α), transforming growth factor beta (TGF-β), interferon gamma (IFN-γ), interleukin-1 (IL-1), and IL-6 [41–43]. Some cytokines stimulate SphK1 activation, followed by the activation of RAS [44] and ERK1/2 and the transcription of NF-κB [45], resulting in inflammation. The inflammatory response to malaria destroys the *Plasmodium* parasite, but negative effects also occur, including cell membrane damage, especially in endothelial cells. Therefore, the expression of S1PRs on the cell surface was dramatically decreased in response to cell injury or cell death. This result suggested that S1PRs can be downregulated by cell injury, which might be stimulated by proinflammatory cytokines.

In addition, the current study demonstrated an increase in SphK1 in malaria-infected mice, which was consistent with previous reports on inflammatory conditions and cancer [9, 14, 15, 46–48]. This finding may stem from inflammation-induced apoptosis. The potential role of SphK1 expression in this model might be the consequence of malaria infection. After malaria infection, the parasites travel to the liver, where they replicate to generate many schizonts in hepatocytes. This situation induces hepatocyte rupture, followed by merozoite release into circulation [2]. In addition, these merozoites can infect red blood cells, where they replicate and develop into the sexual form [2]. As a result, immunological responses are triggered, including the release of proinflammatory cytokines and white blood cell accumulation [30, 49, 50]. TNF-α and other proinflammatory cytokines (IL-1 and IL-6) have been shown to activate SphK1, leading to increased S1P synthesis and release [14]. Cytoplasmic S1P can be irreversibly degraded into ethanolamine phosphate and hexadecenal by the intracellular protein sphingosine phosphate lyase [51]. The hexadecenal molecule can form adducts with proteins, lipids, and DNA and inhibit histone deacetylases (HDACs) in the nucleus, thereby further promoting inflammatory responses in the cytoplasm [14]. In response to inflammation, proinflammatory cytokines stimulate sphingomyelin hydrolysis by alkaline sphingomyelinase [14]. This action initiates ceramide production from hepatocyte membranes. The increase in ceramide in hepatocytes leads to oxidative stress, proinflammatory cytokine secretion, and hepatocyte apoptosis [52]. Therefore, these proinflammatory cytokines can stimulate hepatocyte apoptosis, resulting in a decrease in S1PRs during SphK1 expression in malaria (Fig 4). In addition, SphK2 evokes apoptotic signaling by promoting the local production of S1P [10]. Alternately, the binding of released S1P and S1PRs initiates various intracellular downstream signaling pathways, such as the NF-κB, ERK1/2, and Rac/PLC pathways, which can induce cell survival, proliferation, and migration [10, 14]. A previous study reported that apoptosis stimulated SphK1 expression in Jurkat T cells and U937 monocytes [53]. In a final attempt to prevent cell death, a cell might increase the levels of SphK1 and S1P, which are survival mediators. In addition, the activation of intracellular SphK1 has been reported to protect pulmonary vascular endothelial cells

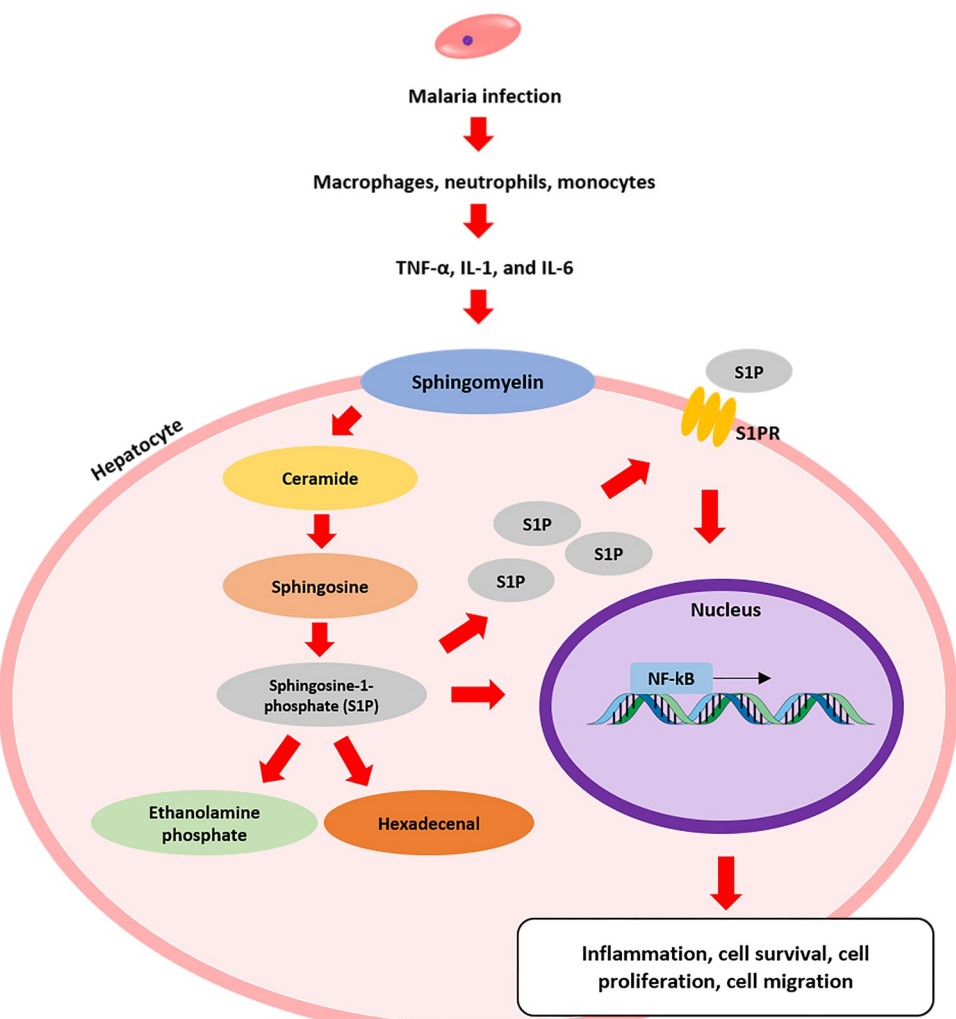

**Fig 4. A schematic illustration of SphK/S1PR signaling during malaria infection.** *Plasmodium* parasite-infected red blood cells induce the accumulation of macrophages, neutrophils, and monocytes, which further initiates the release of proinflammatory cytokines such as TNF-α, IL-1, and IL-6. These proinflammatory cytokines, especially TNF-α, bind to specific receptors on the cell membrane, which can activate sphingomyelin hydrolysis. Next, ceramide can be hydrolyzed to sphingosine by neutral ceramidase. Then, SphKs phosphorylate sphingosine to generate S1P, which directly induces NF-κB activation and translocates to the extracellular space, where it can bind S1PRs and stimulate intracellular downstream signaling, resulting in inflammation and cell survival, proliferation, and migration. In addition, S1P can be degraded into ethanolamine phosphate and hexadecenal; in particular, hexadecenal can stimulate further inflammation in the cytoplasm.

during LPS-mediated inflammation in vivo [54]. This result suggested that SphK1 might play an important role in preventing cell injury under inflammatory conditions. Further studies should focus on the mechanism of SphK and S1PR signaling during liver injury caused by malaria infection.

Some limitations of this study should be noted. The duration of SphK and S1PR changes during malaria infection could not be determined. Therefore, further kinetic studies on SphK and S1PR expression are required to characterize the time-dependent expression of these proteins. Second, the protein levels of SphKs in this animal model were not assessed. Thus, these protein levels should be monitored in future experiments.

## Conclusion

The present study indicated that liver tissue from malaria-infected mice has decreased expression of S1PR1, S1PR2, and S1PR3 and increased expression of SphK1. These findings suggest that alterations in S1PR1, S1PR2, S1PR3 and SphK1 are involved in the inflammatory response and apoptosis and play important roles in the pathogenesis of malaria in the liver.

## Acknowledgments

The authors would like to acknowledge the laboratory workers at the Animal Experiment Building, Walailak University, Thailand, for facilitating all procedures performed in mice. We also thank the staff members at the Department of Tropical Pathology, Faculty of Tropical Medicine, Mahidol University, Thailand, for helping with histological processing.

## Author Contributions

**Conceptualization:** Tachpon Techarang, Pitchanee Jariyapong, Chuchard Punsawad.

**Data curation:** Tachpon Techarang, Pitchanee Jariyapong, Chuchard Punsawad.

**Formal analysis:** Tachpon Techarang, Pitchanee Jariyapong, Chuchard Punsawad.

**Funding acquisition:** Chuchard Punsawad.

**Investigation:** Tachpon Techarang, Pitchanee Jariyapong, Chuchard Punsawad.

**Methodology:** Tachpon Techarang, Pitchanee Jariyapong, Chuchard Punsawad.

**Project administration:** Chuchard Punsawad.

**Supervision:** Chuchard Punsawad.

**Validation:** Tachpon Techarang, Pitchanee Jariyapong, Chuchard Punsawad.

**Writing – original draft:** Tachpon Techarang, Pitchanee Jariyapong, Chuchard Punsawad.

**Writing – review & editing:** Pitchanee Jariyapong, Chuchard Punsawad.

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
