## [Decision Letter · Decision Letter 0]

22 Dec 2021

PONE-D-21-35173Role of sphingosine kinase and sphingosine-1-phosphate receptor on liver pathology in mice infected with Plasmodium berghei ANKAPLOS ONE

Dear Dr. Punsawad,

Thank you for submitting your manuscript to PLOS ONE. After careful consideration, we feel that it has merit but does not fully meet PLOS ONE’s publication criteria as it currently stands. Therefore, we invite you to submit a revised version of the manuscript that addresses the points raised during the review process.

We look forward to receiving your revised manuscript.

Kind regards,

Olga A Sukocheva, PhD

Academic Editor

PLOS ONE

Journal Requirements:

2.  To comply with PLOS ONE submissions requirements, in your Methods section, please provide additional information on the animal research and ensure you have included details on methods of sacrifice.

4. Thank you for stating the following in the Funding Section of your manuscript: 

(This work was partially supported by Walailak University, Thailand. The funders had no role in study design, data collection and analysis, decision to publish, or preparation of the manuscript.)

(NO)

Additional Editor Comments:

Sound study with interesting findings. However, Introduction and Discussion section should be improved; reference list should be extended and present recent relevant reviews in the field of Sphingosine kinase signaling role during activation of inflammation. All Reviewers' suggestions should be addressed. Immunohistopathology images with higher resolution or magnification should be presented. Suggested/ potential S1P-receptor -targeting agents/or other methods for malaria treatment should be indicated in the Discussion section.

Reviewers' comments:

Reviewer's Responses to Questions

**Comments to the Author**

1. Is the manuscript technically sound, and do the data support the conclusions?

Reviewer #1: Partly

Reviewer #2: Yes

2. Has the statistical analysis been performed appropriately and rigorously? 

Reviewer #1: Yes

Reviewer #2: Yes

3. Have the authors made all data underlying the findings in their manuscript fully available?

Reviewer #1: Yes

Reviewer #2: Yes

4. Is the manuscript presented in an intelligible fashion and written in standard English?

Reviewer #1: Yes

Reviewer #2: Yes

5. Review Comments to the Author

Reviewer #1: The manuscript is interesting; however, there are a few issues that need to be corrected before the manuscript is considered for publication.

1. The main issue is with animal experiment execution. Why were control mice not injected with red blood cells but just saline? This is not a proper control as authors introduce additional variables!

2. The method of erythrocyte isolation, infection, etc., has not been described (and is not included in any of the referenced papers). Moreover, one of the referenced manuscripts on page 96 is a case study of an infected patient and does not include any animal experiment. What is the rationale for referencing it here?

3. Not all abbreviations are adequately explained (e.g., line 110) – please correct.

4. Figure 2 is of low quality, and it is impossible to evaluate the staining results.

5. More appropriate for expression data would be boxplot and not the histogram. Also, Figure 3 is incorrectly referenced in the text as Figure 2.

6. Line 208: Authors contradict themselves. First, they write that there are differences in mRNA level of SphK1, to add later that it did not change… Moreover, S1PR1 and S1PR2 mRNA expression are lower only by 30-40% when compared with appropriate controls, so it is not a 3-fold difference. Not to mention, these results are not statistically significant due to high variability between individual animals.

7. Line 251. Downregulation of mRNA expression was only shown for S1PR3, so writing that it affects all investigated receptors is an overstatement.

8. It is not unusual to not see differences at the mRNA level but notice the differences at the protein level (and vice versa), and authors should discuss this in their discussion. Moreover, it would be beneficial for the readers if possible explanation or more detailed information about receptors processing/degradation could be included in the discussion.

Reviewer #2: The study tested levels of SphK1 mRNA expression in liver tissues of mice infected with malaria pathogens. Authors also found that S1PR1, S1PR2, and S1PR3 expressions are decreased in the liver tissue of Plasmodium berghei-infected mice. Data suggests that the decreased levels of S1PR1, S1PR2, and S1PR3 may be involved in liver injury during malaria infection. The study is very interesting. However, there are several issues to address. The manuscript should be improved/amended.

1. Text requires proper English language editing.

2. Figure 2 resolution is poor. Author should provide images with higher magnification and/or better resolution.

3. Introduction/discussion section sections do not provide enough information; several important citations are missing. Role of SPhK in the infection resolution should be covered in more details ( see these reviews: https://pubmed.ncbi.nlm.nih.gov/31863815/ ; https://pubmed.ncbi.nlm.nih.gov/28075451/

https://pubmed.ncbi.nlm.nih.gov/29385066/

4. Was SphK1 protein level measured/compared? Authors should discuss SPhK1 protein expression and mechanisms of its regulation considering the activation of receptor degradation during inflammation.

5. S1P receptors intracellular processing was not discussed ( see this report https://pubmed.ncbi.nlm.nih.gov/23142484/)

6. The manuscript will benefit if authors include schematic signaling pathway for SPhK/S1P receptor signalling in the infected/normal liver.

7. Study limitations section should be extended.

6. PLOS authors have the option to publish the peer review history of their article (what does this mean?). If published, this will include your full peer review and any attached files.

Reviewer #1: No

Reviewer #2: No

---

## [Author Response · Author response to Decision Letter 0]

3 Mar 2022

POINT-BY-POINT RESPONSES TO THE REVIEWERS’ COMMENTS

Additional Editor Comments:

Sound study with interesting findings. However, Introduction and Discussion section should be improved; reference list should be extended and present recent relevant reviews in the field of Sphingosine kinase signaling role during activation of inflammation. All Reviewers' suggestions should be addressed. Immunohistopathology images with higher resolution or magnification should be presented. Suggested/ potential S1P-receptor -targeting agents/or other methods for malaria treatment should be indicated in the Discussion section.

Response: We has been revised as suggested.

Review Comments 

Reviewer #1: The manuscript is interesting; however, there are a few issues that need to be corrected before the manuscript is considered for publication.

1. The main issue is with animal experiment execution. Why were control mice not injected with red blood cells but just saline? This is not a proper control as authors introduce additional variables!

Response: In the animal model, Plasmodium berghei ANKA-infected red blood cells were diluted to 1×106 cells/ml with saline solution as a diluent before intraperitoneal injection [1]. Therefore, the control mice were injected with the same volume of diluent (saline solution).

2. The method of erythrocyte isolation, infection, etc., has not been described (and is not included in any of the referenced papers). Moreover, one of the referenced manuscripts on page 96 is a case study of an infected patient and does not include any animal experiment. What is the rationale for referencing it here?

Response: We apologize for our mistake regarding the reference. In response to this comment, the reference was changed, and malaria parasite inoculation and infected red blood cell collection were described in more detail. Please see page 5, lines 96-100.

3. Not all abbreviations are adequately explained (e.g., line 110) – please correct.

Response: Thank you for this valuable suggestion. This abbreviation refers to the names and surnames of two expert pathologists. In response to this comment and to avoid confusion, the abbreviations have been deleted. Please see page 6, line 114.

4. Figure 2 is of low quality, and it is impossible to evaluate the staining results.

Response: Figure 2 has been revised as suggested. The new image has a resolution of 600 dpi. Please see the attached file.

5. More appropriate for expression data would be boxplot and not the histogram. Also, Figure 3 is incorrectly referenced in the text as Figure 2.

Response: Thank you very much for your comment. In response to this comment, we have revised the graph and corrected the reference as suggested. Please see Figure 3 and page 11 (lines 212 and 214).

6. Line 208: Authors contradict themselves. First, they write that there are differences in mRNA level of SphK1, to add later that it did not change… Moreover, S1PR1 and S1PR2 mRNA expression are lower only by 30-40% when compared with appropriate controls, so it is not a 3-fold difference. Not to mention, these results are not statistically significant due to high variability between individual animals.

Response: First, we apologize for the error that you noted. In the later instance mentioned regarding SphK1, we were referring to SphK2 mRNA levels. Therefore, we altered the text to indicate the correct marker in response to this criticism. Please see page 11, line 211. Second, we double-checked and reanalyzed the laboratory results. As you mentioned, we discovered that the mRNA expression levels of S1PR1 and S1PR2 were reduced by 30–40% in the PbA-infected group. However, a significant decrease in S1PR3 mRNA expression was observed in the PbA-infected group compared with the control group. As a result, the text and graph have been updated. Please see lines 211–214 on page 11 and Figure 3.

7. Line 251. Downregulation of mRNA expression was only shown for S1PR3, so writing that it affects all investigated receptors is an overstatement.

Response: In response to comment 6, we double-checked the results. Except for S1PR3, all the receptors had lower mRNA expression in the PbA-infected group, but the differences were not significant.

8. It is not unusual to not see differences at the mRNA level but notice the differences at the protein level (and vice versa), and authors should discuss this in their discussion. Moreover, it would be beneficial for the readers if possible explanation or more detailed information about receptors processing/degradation could be included in the discussion.

Response: We have updated the discussion on the interaction between SphKs and S1PRs. Please see page 15, lines 293–298.

Reviewer #2: The study tested levels of SphK1 mRNA expression in liver tissues of mice infected with malaria pathogens. Authors also found that S1PR1, S1PR2, and S1PR3 expressions are decreased in the liver tissue of Plasmodium berghei-infected mice. Data suggests that the decreased levels of S1PR1, S1PR2, and S1PR3 may be involved in liver injury during malaria infection. The study is very interesting. However, there are several issues to address. The manuscript should be improved/amended.

1. Text requires proper English language editing.

Response: A revised manuscript was thoroughly edited by highly qualified native English-speaking editors at American Journal Experts.

2. Figure 2 resolution is poor. Author should provide images with higher magnification and/or better resolution.

Response: Figure 2 has been revised as suggested. The new image has a resolution of 600 dpi. Please see the attached file.

3. Introduction/discussion section sections do not provide enough information; several important citations are missing. Role of SPhK in the infection resolution should be covered in more details (see these reviews: https://pubmed.ncbi.nlm.nih.gov/31863815/ ; https://pubmed.ncbi.nlm.nih.gov/28075451/

https://pubmed.ncbi.nlm.nih.gov/29385066/

Response: Thank you very much for this valuable suggestion. In response to this comment, we have added more information on the role of SphKs in the resolution of infection in the Introduction section. Please see page 4, lines 69–72.

4. Was SphK1 protein level measured/compared? Authors should discuss SPhK1 protein expression and mechanisms of its regulation considering the activation of receptor degradation during inflammation.

Response: SphK1 protein levels were not measured. However, we qualitatively evaluated SphK and S1PR proteins in the liver by using an immunohistochemistry assay. In response to this comment, text on SphK1 protein expression and activity during inflammation has been added to the Discussion section. Please see pages 14–15, lines 278–298.

5. S1P receptors intracellular processing was not discussed (see this report https://pubmed.ncbi.nlm.nih.gov/23142484/)

Response: We have discussed the interaction between SphKs and S1PRs. Please see page 15, lines 293–298.

6. The manuscript will benefit if authors include schematic signaling pathway for SPhK/S1P receptor signalling in the infected/normal liver.

Response: Thank you very much for this valuable suggestion. A schematic illustration of the mechanism identified in this study has been added to the Discussion section. Please see pages 15, lines 313–323, and the attached file.

7. Study limitations section should be extended.

Response: The study limitations have been updated. Please see page 15, lines 309–311.

---

## [Decision Letter · Decision Letter 1]

14 Mar 2022

Role of sphingosine kinase and sphingosine-1-phosphate receptor in the liver pathology of mice infected with Plasmodium berghei ANKA

PONE-D-21-35173R1

Dear Dr. Punsawad,

We’re pleased to inform you that your manuscript has been judged scientifically suitable for publication and will be formally accepted for publication once it meets all outstanding technical requirements.

Kind regards,

Olga A Sukocheva, PhD

Academic Editor

PLOS ONE

Additional Editor Comments (optional):

Reviewers' comments:

Reviewer's Responses to Questions

**Comments to the Author**

1. If the authors have adequately addressed your comments raised in a previous round of review and you feel that this manuscript is now acceptable for publication, you may indicate that here to bypass the “Comments to the Author” section, enter your conflict of interest statement in the “Confidential to Editor” section, and submit your "Accept" recommendation.

Reviewer #1: All comments have been addressed

Reviewer #2: All comments have been addressed

2. Is the manuscript technically sound, and do the data support the conclusions?

Reviewer #1: Yes

Reviewer #2: Yes

3. Has the statistical analysis been performed appropriately and rigorously? 

Reviewer #1: Yes

Reviewer #2: Yes

4. Have the authors made all data underlying the findings in their manuscript fully available?

Reviewer #1: Yes

Reviewer #2: Yes

5. Is the manuscript presented in an intelligible fashion and written in standard English?

Reviewer #1: Yes

Reviewer #2: Yes

6. Review Comments to the Author

Reviewer #1: (No Response)

Reviewer #2: The revised version of the manuscript has been improved. Authors addressed all my comments properly.

7. PLOS authors have the option to publish the peer review history of their article (what does this mean?). If published, this will include your full peer review and any attached files.

Reviewer #1: No

Reviewer #2: No

---

## [Editor Report · Acceptance letter]

18 Mar 2022

PONE-D-21-35173R1 

Role of sphingosine kinase and sphingosine-1-phosphate receptor in the liver pathology of mice infected with *Plasmodium berghei* ANKA 

Dear Dr. Punsawad:

I'm pleased to inform you that your manuscript has been deemed suitable for publication in PLOS ONE. Congratulations! Your manuscript is now with our production department. 

Kind regards, 

on behalf of

Dr. Olga A Sukocheva 

Academic Editor

PLOS ONE